# Clinical Strategies Targeting the Tumor Microenvironment of Pancreatic Ductal Adenocarcinoma

**DOI:** 10.3390/cancers14174209

**Published:** 2022-08-30

**Authors:** Nebojsa Skorupan, Mayrel Palestino Dominguez, Samuel L. Ricci, Christine Alewine

**Affiliations:** 1Laboratory of Molecular Biology, National Cancer Institute, National Institutes of Health, Bethesda, MD 20892, USA; 2Medical Oncology Program, Center for Cancer Research, National Cancer Institute, National Institutes of Health, Bethesda, MD 20892, USA

**Keywords:** immunotherapy, pancreatic cancer, stromal modifiers, cancer-associated fibroblasts

## Abstract

**Simple Summary:**

Tumors from pancreatic cancer contain many types of cells (such as immune cells and fibroblasts) in addition to the cancer cells. Using targeted drugs to disrupt interactions between these cells which can support cancer cell growth, invasion, and immune suppression has become an important area of exploration in the pancreatic cancer field. This review describes new drugs designed to modulate interactions between cancer cells and other cell types in the tumor and discusses the initial clinical trials testing these novel therapeutics in pancreatic cancer patients.

**Abstract:**

Pancreatic cancer has a complex tumor microenvironment which engages in extensive crosstalk between cancer cells, cancer-associated fibroblasts, and immune cells. Many of these interactions contribute to tumor resistance to anti-cancer therapies. Here, new therapeutic strategies designed to modulate the cancer-associated fibroblast and immune compartments of pancreatic ductal adenocarcinomas are described and clinical trials of novel therapeutics are discussed. Continued advances in our understanding of the pancreatic cancer tumor microenvironment are generating stromal and immune-modulating therapeutics that may improve patient responses to anti-tumor treatment.

## 1. Introduction

Pancreatic cancer is a highly aggressive malignancy with a 5-year overall survival of only 10% despite advances in systemic therapy over the last decade [1,2,3]. While pancreatic cancer represents just 3.2% of new cancer diagnoses, given its high mortality, it has become the third most common cause of cancer-related death in the United States, and the incidence is rising [4]. Pancreatic ductal adenocarcinoma (PDAC) is the most common histology of pancreatic cancer, representing >85% of all pancreatic cancer diagnoses [5]. Initial clinical symptoms of PDAC are commonly non-specific, which leads to diagnosis at already incurable, advanced stages. Current standard of care for locally advanced or metastatic PDAC consists of combination chemotherapy with FOLFIRINOX or gemcitabine/nanoalbumin-bound (NAB-) paclitaxel (GN) and offers only a few months of overall survival benefit to the fit patients able to tolerate it [1,3].

The precision oncology revolution has largely excluded PDAC. Somatic mutations occur at incidences over 30% in just four genes: *KRAS*, *TP53*, *CDNK2A* and *SMAD4* [6,7]. *KRAS* mutation is the primary oncogenic driver of PDAC and occurs in more than 90% of patient tumors. This target has historically been considered “undruggable” by pharma [8], but recent breakthroughs have led to the development of exciting new small molecule inhibitors [9,10]. Sotorasib, which targets the KRAS G12C mutant, has already been approved in lung cancer [11], and has demonstrated some activity in the 1–2% of PDAC patients with this mutation [12,13]. Patients and providers are still awaiting the arrival of inhibitors for KRAS G12D and G12V mutants, which together account for ~72% of *KRAS* mutations seen in PDAC [8]. By contrast, drugs which can correct the loss of tumor suppressors *TP53*, *CDKN2A* and *SMAD4* remain elusive. Approximately 5–9% of PDAC tumors contain germline or somatic mutation in the DNA-repair-related genes *BRCA1*, *BRCA2*, and *PALB2* [14,15]. Tumors with these mutations are exquisitely sensitive to platinum chemotherapy [16], and treatment with these regimens can more than double patient survival [17,18,19]. These patients may also benefit from targeted treatment with the PARP inhibitor olaparib [20]. Other mutations that can be matched to existing targeted therapies are uncommon in PDAC, and recent efforts to sequence patient tumors in real time and deliver genetically tailored regimens have provided only a few months of clinical benefit, even when it was feasible to obtain and administer these therapies [21,22]. The lower success rate of mutation-matched therapies in PDAC compared to other tumor types is frequently attributed to the concurrent activation of KRAS or to the unique PDAC tumor microenvironment (TME).

PDAC is also largely non-responsive to immunotherapy [23,24]. Compared to most solid tumors, PDAC has a very low mutational burden, a typical marker for immune “cold” tumors. High microsatellite instability (MSI-H) and mismatch repair deficiency (dMMR), markers predictive of response to immune checkpoint inhibitors, occur in less than 2% of PDAC cases [25,26]. However, even in PDAC tumors which bear these genetic changes, response to immune checkpoint inhibitors occurs less frequently than for patients with other types of MSI-H/dMMR tumors [27,28]. It is widely believed that the desmoplastic reaction produced by activated cancer-associated fibroblasts (CAFs) in the PDAC microenvironment creates a tumor niche that is physically difficult for effector leukocytes to access and rich in immunosuppressive chemical signals that cause exhaustion or inactivation if they do successfully breach that barrier [29,30,31].

The proliferation of basic and translational research delineating the signals responsible for the establishment and maintenance of the hostile PDAC TME has permitted the development of highly specific TME-directed therapeutics that are now being tested in PDAC patients. New drugs designed to target TME cellular and extracellular matrix (ECM) components or to block chemical crosstalk between cancer cells, TME fibroblasts, and immune components represent a departure from classic therapeutic strategies aimed at the cancer cells themselves, such as chemotherapeutic poisons or inhibitors blocking the activity of oncogenic drivers. The purpose of this review is to provide an update for physicians and laboratory scientists on the clinical progress of novel therapeutics purported to target and remodel the PDAC TME. Strategies currently under evaluation or recently tested in the clinic which aim to overcome PDAC immunosuppression and treatment resistance are discussed. The number of such strategies has ballooned over the last decade and includes those identified specifically in pre-clinical or translational studies of PDAC and also some that have shown promise for enhancing the activity of immunotherapy or chemotherapy in other tumor types. Here, we focused on biologics (including monoclonal antibodies) and small molecules that can be administered off-the-shelf to patients to manipulate TME cell populations and paracrine signaling in PDAC. Numerous unanswered questions remain about how to best utilize these therapeutics, many of which have little to no single agent anti-tumor activity and may benefit patients only when prescribed with the right combination of other drugs. Nevertheless, new strategies are clearly needed for tackling PDAC. Recombining existing chemotherapy drugs to produce a stronger regimen (i.e., FOLFIRINOX [1,2]) or engineering new formats for old drugs that increase their activity against PDAC (i.e., NAB- paclitaxel [3], irinotecan, liposome (nal-iri) [32]) are our largest therapeutic breakthroughs in this disease over the last fifteen years. These advances have provided incremental benefits for patients with metastatic disease and additional cures in the early-stage setting; however, the vast majority of PDAC patients continue to lack treatment options capable of changing the grim natural history of their disease. TME-modifying agents offer a new chance of overcoming the substantial defenses that PDAC employs to survive our best attempts to eliminate it.

## 2. TME in PDAC

The TME of PDAC contains numerous cell types and exhibits a resilient and exuberant desmoplastic reaction (Figure 1). Desmoplasia generates a dense ECM composed of collagen, fibronectin, laminin, and hyaluronic acid that is synthesized by cancer-associated fibroblasts (CAFs). The resulting fibrosis generates interstitial fluid pressures than can exceed mean arterial pressures. This precipitates vascular compression and creates a considerable barrier to therapeutics that must exit the circulation and reach the vicinity of cancer cells [33]. Several subtypes of CAFs with unique gene signatures and specialized functions have been identified: myofibroblastic CAFs (myCAFs), inflammatory CAFs (iCAFs), and antigen-presenting CAFs (apCAFs) [34,35,36]. Initially, it was thought that all CAF populations arose from pancreatic stellate cells (PSCs) and differentiated to the various subtypes under the influence of environmental cues within the TME, including cytokine and growth factor gradients. However, it has recently been demonstrated that PSC-derived CAFs give rise only to a minor population of myCAFs [35] and that alternative cell populations differentiate into apCAFs [37]. CAFs release cytokines, chemokines, adhesion molecules, and growth factors that influence immune and endothelial cells. CAFs also regulate ECM components that can contribute to metastatic spread and tumor aggressiveness [38,39].

Amongst the cell types that differentiate into PDAC CAFs, the role of PSC-derived CAFs has been extensively studied and has revealed both a tumor-promoting and tumor-restraining role for these cells and their products [29,35,40]. The activation of PSCs results in their transformation from a quiescent to a myofibroblast-like phenotype that expresses high levels of alpha smooth muscle actin (αSMA). In patients with early-stage (T1–T2) PDAC, moderate-to-strong αSMA expression was associated with poorer clinical outcomes compared to tumors with lower levels of αSMA expression [41]. PSC-derived CAFs can inhibit cancer cell apoptosis and promote chemoresistance and disease recurrence. PSCs can accompany cancer cells to distant metastatic sites and help to establish a supportive niche for their growth [42]. In fact, evidence supports that survival of tumor cells in the inhospitable PDAC TME depends on CAFs. Acute destruction of FAP+ (fibroblast-expressing activation protein) CAFs caused rapid hypoxic necrosis of cancer and stromal cells that is interferon-γ (IFN-γ)- and tumor necrosis factor-α (TNF-α)-dependent [43]. Others have delineated a role for FAP(+)-CAFs in immunosuppression [44]. Activated PSCs secrete a dense ECM that modulates tumor stiffness and invasiveness through increased expression of the type IV collagenase matrix metalloproteinase-2 (MMP-2) [40]. Meflin, a glycosylphosphatidylinositol-anchored protein in CAFs, interacts with lysyl oxidase to inhibit collagen crosslinking activity and reduce tissue stiffness. The induced expression of meflin by both genetic and pharmacological approaches increased tumor vessel area in murine PDAC, improved drug delivery, and increased tumor chemosensitivity [45]. This is one of many pre-clinical studies showing that decreased ECM stiffness enhances therapeutic efficacy [33,46]. From these data, one might conclude that elimination of CAFs and reduction in ECM should inhibit PDAC and improve patient outcomes. However, negative clinical studies using Sonic Hedgehog (Hh) inhibitors to ablate CAFs in PDAC patients required the field to reconsider the role of this cell type in PDAC [47,48]. Subsequently, it was found that, paradoxically, depletion of type I collagen from PDAC-bearing mouse stroma significantly decreased animal survival [49]. Furthermore, ablation of αSMA(+) fibroblasts resulted in highly hypoxic, undifferentiated tumors with a more aggressive phenotype [50]. Similarly, inhibition of collagen crosslinking by LOXL2 increased PDAC growth and reduced overall survival [51]. These studies demonstrate that while there is clearly a sub-population of CAFs that facilitates cancer growth, some portion of the CAF population also appears to have a tumor-restraining role.

In the last few years, research into CAFs and their diverse sub-populations has provided increased insight. Nevertheless, correlating individual CAF subtypes with a universally *bad* (tumor-promoting) or *good* (tumor-restraining) phenotype is more difficult. The literature is generally concordant in condemning iCAFs as *bad* actors in the PDAC TME. These cells have been implicated in neoplastic progression to PDAC through induction of inflammation and complement regulatory factors [52]. Moreover, iCAFs can aid the preservation of cancer stem cells and facilitate chemotherapy resistance by modulating TME metabolism [53]. Conversely, myCAFs’ effect on cancer cells appears variable and situational. Depletion of myCAFs by Hh inhibition changes the balance of T cell subsets to generate a more immunosuppressive tumor, while at the same time impairing tumor growth, at least in the short-term setting [54]. It appears that apCAFs could fall on the *good* side in some contexts; these cells can induce CD25 and CD69 immune activators in co-cultured T cells [34]. However, there is evidence showing that apCAFs differentiated from mesothelial cells transform CD4^+^ T cells into Treg cells in mice [37]. Adding to the complexity, it has been suggested that iCAF and myCAF determination is not static, but that cells could be “interconvertible between types” under the influence of the correct chemical signals. In fact, treatment of tumors with inhibitors of the JAK/STAT pathway, a critical agent in iCAF differentiation, resulted in an increased myCAF to iCAF ratio [55]. Recently, a fourth type of CAF, highly activated metabolic state CAF (meCAF), was identified by single-cell analysis of PDAC specimens of varying desmoplastic exuberance. These meCAFs were highly prevalent in low density tumors and predictive of poorer prognosis. However, their presence also predicted better response to GN/anti-programmed cell death protein 1 (PD-1) blockade in PDAC patients, an effect attributed to enhanced immune surveillance compared to tumors with high desmoplasia [56]. The varying roles of CAFs make pharmacological targeting of CAFs and ECM components highly complex, since proposed strategies must selectively eliminate tumor-promoting elements without inhibiting or eradicating tumor-restraining ones. Cautious selection of targets after rigorous pre-clinical testing is necessary to offer patients the least chance of unintentional harm.

## 3. PDAC Is Defined by an Immunosuppressive TME

Mutations in the *KRAS* oncogene are present in >90% of PDAC patients and activation of this pathway defines the disease [57,58,59]. Mutated KRAS in combination with inflammation or loss of key tumor suppressors drives progression of pre-malignant lesions to PDAC and is implicated in the recruitment of an immunosuppressive cellular milieu through KRAS-driven production of pro-inflammatory cytokines and chemokines [60,61,62,63]. Oncogenic KRAS has also been implicated in tumor immune evasion [64,65]. The role of mutated KRAS in establishing the immunosuppressive PDAC TME has been well described by others [66,67,68].

PDAC tumors have robust infiltration by T cells. Unfortunately, most of these cells promote tumorigenesis; cytotoxic T cells are infrequent in the PDAC TME. The most abundant T cell subtype is CD4^+^ regulatory T (Treg) cells. Tregs play a crucial role in warding off the host immune system. Tregs are increased in PDAC, conferring poor prognosis in patients. Several depletion experiments established Tregs to be suppressors of anti-tumor immune responses [69]. Interestingly, most cytotoxic CD8^+^ T cells are excluded from the vicinity of pancreatic cancer cells. In patients, the spatial proximity of cytotoxic CD8^+^ T cells, but not CD4^+^ T cells or total T cells, to pancreatic cancer cells correlates with increased overall survival [70].

The majority of immune cells in the TME are of myeloid origin. These include tumor-associated macrophages (TAMs), granulocytes, and inflammatory monocytes. During pre-cancerous stages, activated KRAS actively recruits these cells to the TME. TAMs are some of the most abundant immune cells in PDAC and their multiple roles have been extensively described previously [71]. TAMs can generally be categorized as either M1 or M2 polarized. M1 macrophages are considered tumor suppressive. When activated, they secrete TNF-α, interleukin (IL)-12, IL-1α, and IFN-γ, which can have a tumoricidal effect. Conversely, M2 macrophages are generally immunosuppressive. Their products, such as transforming growth factor-beta (TGF-β) and IL-10, tend to be tumor-promoting [72]. TAMs do not operate in isolation. TAMs and collagen in the TME interact to shape each other. High collagen density, as found in PDAC, promotes an immunosuppressive macrophage phenotype [73]. At the same time, TAMs can internalize collagen matrix through the action of the mannose receptor (MRC1), resulting in increased arginine synthesis from biproducts of lysosomal collagen breakdown. The high levels of arginine result in increased production of reactive nitrogen species, which in turn promote a profibrotic phenotype in PSCs. This leads to increased fibrosis and formation of more collagen [74]. TAMs can also affect cancer cell programming. In fact, the presence of TNF-α-secreting macrophages can push cancer cells to take on a more aggressive basal-like subtype [75]. Additionally, circulating monocytes and TAMs contribute to the development of the pre-metastatic niche by activating resident hepatic stellate cells. This promotes a fibrotic microenvironment that sustains metastatic tumor growth [76]. TAMs serve many roles in the PDAC TME.

MDSCs (myeloid-derived suppressor cells) are also derived from myeloid cells. They can be subtyped into monocytic or granulocytic and are known to exert immunosuppressive effects on T cells via arginase, nitric oxide synthase, TGF-β, IL-10, and COX2. MDSCs are recruited early in the process of carcinogenesis and promote the formation and maintenance of pre-neoplastic lesions. MDSCs also recruit Tregs to the TME. A more precise understanding of MDSCs has been difficult to achieve given their heterogeneity in both mice and humans [77].

Neutrophils are essential infiltrating immune cells in the PDAC TME, but tumor-associated neutrophils (TANs) are less mechanistically established in pancreatic carcinogenesis as compared with TAMs. TANs are detected even at early stages of the PDAC development. In mice, knockout of CXCR2, the primary neutrophil chemotaxis receptor, inhibits TAN infiltration into tumors, leading to T cell-dependent tumor growth inhibition [30]. Some studies classify TANs into two polarization states, tumor-suppressing N1 neutrophils and tumor-promoting N2 neutrophils [78]. Pro-inflammatory or immunostimulatory cytokines, such as IL-12, CXCL9, CXCL10, and CCL3, are released from N1 neutrophils and facilitate recruitment and activation of CD8^+^ T cells. On the other hand, exposure to TGF-β transforms neutrophils to the N2 phenotype [78]. N2 neutrophils have been reported to have strong immunosuppressive and tumor-supporting functions, including the promotion of tumor metastases and angiogenesis. Poor patient outcomes are associated with high intratumoral neutrophils in advanced cancer patients [79]. 

Dendritic cells (DCs) are professional antigen-presenting cells (APCs) that participate in both innate and adaptive immune responses and are critical to boosting immune responses to antigens, including tumor-associated antigens. DC responses are impaired in patients with PDAC. Specifically, PDAC secretes cytokines such as IL-6, IL-10, and TGF-β, which reduce the stimulatory capacity of DCs [80]. Overall numbers of circulating DCs are also noted to be lower in PDAC patients [81,82]. Interestingly, function and numbers of circulating DCs rebound following surgical resection of tumors. Within PDAC tumors, conventional DCs (cDCs) are largely excluded, a process which appears to begin in pre-malignant pancreas lesions [83,84]. In mice, restoration of cDC populations to established tumors is alone insufficient to break immune tolerance and regress tumors; stimulation to overcome low DC function is also required, including the presence of tumor antigens [84]. Despite this, increased numbers of circulating DCs and tumor DCs have been associated with better survival in patients with pancreatic cancer [85,86].

The role of B cells in PDAC tumorigenesis remains controversial. While B cells clustered in tertiary lymphoid structures are associated with better outcome in PDAC patients, manipulations that increase B cells have negative impacts on survival in several PDAC mouse models [87]. Most B cells are considered pro-inflammatory and immune-stimulatory, while ~10% are immunosuppressive. Under hypoxic conditions, CXCL13 secreted by CAFs attracts immunosuppressive B cells to tumors [88], which can then induce M2 polarization of TAMs [89]. In addition, B cells can activate CAFs via the soluble factor platelet derived growth factor-B (PDGF-B) [90,91]. It is difficult to reconcile the pro-tumor role of B cells in mice with developing tumors with the human data from well-established tumors. This apparent paradox suggests that B cells play different roles as the tumor progresses and evolves.

Natural killer (NK) cells are defined by the lack of surface T cell receptors (TCRs), the expression of the neural cell adhesion molecule (NCAM), and the natural cytotoxicity receptor (NCR) NKp46. NK cells recognize and directly kill virus-infected or tumor cells without prior antigen stimulation [92]. NK cells can kill by multiple mechanisms, including secretion of perforin to destroy cell membranes, release of granzymes for a lytic killing effect, or activation of the Fas/FasL pathway to induce apoptosis of target cells. Activated NK cells also secrete cytokines, such as TNF-α, IFN-γ, and granulocyte-macrophage colony-stimulating factor (GM-CSF), which trigger activation and recruitment of other innate and adaptive immune cells that broaden and strengthen the anti-tumor immune response [93]. For example, IFN-γ secretion by NK cells is critical in shaping T cell responses, including T_H_1 polarization and CD8^+^ T cell activation [94]. PDAC patients have normal numbers of peripheral NK cells, but NK-cell activity is progressively impaired at more advanced stages of disease. NK activating receptors NKG2D and NKp30 are expressed at lower levels in PDAC patients [95]. In addition, these cells have decreased cytotoxic activity, low IFN-γ expression, and high intracellular levels of IL-10. Within tumor tissue, NK cells are largely excluded and display a decreased activity and toxicity potential [96]. In PDAC, several mechanisms impair the NK cell function and polarize NK cells towards a tumor-promoting phenotype. Cancer cells suppress NK cell function through expression of TGF-β, IL-10, indoleamine 2,3-dioxygenase (IDO), and matrix metalloproteinases (MMPs), which impair NK cell tumor recognition and killing via the downregulation of cytotoxicity receptors. Another mechanism of NK-cell inhibition is the secretion of the Igγ-1 chain C region (IGHG1), which competitively binds to the Fcγ receptor of NK cells, reducing antibody-dependent cell-mediated cytotoxicity (ADCC). Both cancer cells and PSC-derived CAFs secrete IL-18, IL-10, and TGF-β, all of which diminish NK-cell function [92]. Interestingly, chemotherapy can restore NK-cell-mediated anti-tumor activity of endogenous NK cells in mouse models, an intriguing therapeutic side effect in the age of immunotherapy [97].

## 4. Targeting the PDAC TME with Immune-Modulating Agents

Immunotherapy in PDAC has been extensively reviewed previously [98]. Use of immune checkpoint inhibitors targeting cytotoxic T-lymphocyte-associated protein 4 (CTLA-4), PD-1 and programmed death-ligand 1 (PD-L1) has revolutionized oncology due to their unprecedented levels of activity in tumors such as melanoma [99], non-small cell lung cancer [100,101], renal cell carcinoma [102], and hepatocellular carcinoma [103]. Immune checkpoint inhibitors block binding of tolerogenic ligands CTLA-4 and PD-1 to their cognate receptors, and therefore promote the activity of host anti-tumor T cells silenced by tumor activation of these pathways. Unfortunately, PDAC is almost universally refractory to these immunotherapy agents. Trials of single agent checkpoint inhibitors in PDAC resulted in no responses [23,104], except for the <1% of patients with MSI-H/dMMR tumors [105,106]. Dual treatment with anti-CTLA-4 plus anti-PD-1 or anti-PD-L1 agents resulted in enhanced anti-tumor activity at the expense of increased toxicity for patients with some tumor types such as melanoma [107], but was unsuccessful in PDAC patients [24]. While increased anti-tumor activity was observed by combining immune checkpoint inhibitors with chemotherapy in lung and breast cancers [108,109], these combinations have demonstrated limited activity in PDAC, leading to labeling of PDAC as an immunologically “cold” tumor [110,111,112,113]. Administration of some therapeutic anti-cancer vaccines (±chemotherapy and/or immune checkpoint inhibitor) have been shown to induce a more favorable immune environment; however, improved clinical activity with these agents as compared to standard-of-care treatments has yet to be conclusively demonstrated [114,115,116,117,118,119,120,121,122]. Recently, adoptive cell therapy has met with isolated success in PDAC [123], but extension of its therapeutic benefit to a wider population is likely to require therapeutic combinatorial approaches to overcome barriers to T cell infiltration and sustained cell activation within the hostile, nutrient-poor TME. Current efforts to define an immunotherapy regimen that can benefit PDAC patients involves multimodal strategies to transform the PDAC TME, to enhance endogenous T cell activity, or to increase efficacy of adoptively transferred T cell immunity [124]. Novel therapies designed to block immunosuppressive signals from cancer cells or to directly agonize anti-tumor immunity have entered the clinic and are currently being tested in PDAC patients (Table 1).

### 4.1. CD40 Agonists

#### 4.1.1. Role of CD40 in Immunity and Rationale for Its Use in Cancer Patients

CD40 is a broadly expressed receptor molecule belonging to the TNF superfamily. Binding of the CD40 ligand (CD40L; CD154), expressed on CD4^+^ T cells, stimulates APCs, especially DCs, causing upregulation of APC surface molecules critical for activation of the adaptive immune cascade [125]. Under the influence of CD40, signals from DCs can activate CD8^+^ cytotoxic T cells in the absence of CD4^+^ T cell help, resulting in potent immune stimulation that is independent of T cell checkpoint inhibition [126]. CD40 also has a crucial role in effector T cell maturation; without it CD8^+^ T cells cannot be primed or activated [127]. Interestingly, the deficiencies in DC number and function that have been observed in PDAC can be partially reversed by CD40 stimulation. In fact, in mouse models of PDAC, CD40 activation combined with chemotherapy, radiotherapy, or immune checkpoint inhibition has caused tumor regressions [128,129,130], predominantly via T cell activation [131].

Multiple studies have shown that PDAC patients harbor T cells which recognize tumor antigens [132,133,134]. Inability of these T cells to mount an immune response against PDAC, even in the presence of immune checkpoint inhibitor therapies that prevent T cell exhaustion/tolerance, is at least partially attributed to inadequate help from the defunct DC population [135]. Agonistic CD40 monoclonal antibodies have been under active investigation as novel immunomodulatory agents that could potentially overcome tumor signals that weaken DCs, leading to enhanced antigen-dependent DC activity and a break in tumor immune tolerance. Additionally, CD40 agonism could provide important co-stimulation to enhance the efficacy of anti-cancer vaccines [135]. In pre-clinical models, CD40 agonists mimic CD4^+^ T cell-mediated activation of DCs and induce secretion of T_H_1 cytokines such as IL-12. CD40 agonist stimulation can also induce DC activation and change TAM polarization from an M2- to M1-like phenotype to deplete tumor stroma and provide additional immune attack on tumors [131,136]. These actions make CD40 agonism complimentary to and potentially synergistic with blockade of CTLA-4 and/or PD-(L)1, as it serves to further activate unblockaded T cells. Enhanced activity of immune checkpoint blockade has been seen in pre-clinical studies examining the combination [137,138]. In addition, combination with properly sequenced chemotherapy results in release of tumor antigens that augment the CD40 agonistic effect [139]. Pre-clinical studies suggest that the timing and sequence of when CD40 agonists are delivered in combination regimens can dramatically change their efficacy. Specifically, maneuvers that release tumor antigens (such as administration of chemotherapy) must occur prior to CD40 agonist delivery [139].

#### 4.1.2. Clinical Trials of CD40 Agonists in PDAC Patients

Selicrelumab (CP-870,893; RO7009789) is a fully human IgG2 CD40 agonist monoclonal antibody (mAb) that has been extensively tested for multiple oncology indications including PDAC. A Phase 1 clinical trial of single-agent selicrelumab in patients with advanced solid tumors demonstrated a favorable toxicity profile, and also produced radiologic responses in some melanoma patients [140]. Treatment-related adverse events included dose-related cytokine release syndrome (CRS), transient elevation of serum transaminases, and transient decreases of peripheral lymphocytes, monocytes, and platelets. CRS was mild in most patients and could be managed in the outpatient setting with supportive care. It was associated with elevated serum IL-6 and TNF-α, and manifested rapidly after infusion with symptoms including fever, rigors or chills, rash, back pain, and muscle aches. Patient symptoms generally resolved within 24 h.

Activity of selicrelumab in advanced PDAC patients was tested in a follow-up combination study where the CD40 agonist was administered with standard-of-care gemcitabine (NCT0071191). Treatment tolerability was redemonstrated; the selicrelumab side effect profile was similar to what had been seen in previous studies. Partial responses occurred in 4 of 22 patients, a rate of response higher than what would be expected for single-agent gemcitabine [136,141]. Interestingly, tumor tissue from a responding patient showed no evidence of lymphocyte infiltration, an observation that was subsequently recapitulated in the KPC mouse model. Further testing in mice showed that the anti-tumor effect was dependent on the systemic macrophage population rather than T lymphocytes [136]. To better define the biological effect of CD40 agonism on human PDAC, a window of opportunity study was performed in surgically resectable patients (NCT02588443). Two weeks before surgery, participants received a single dose of either selicrelumab alone or selicrelumab preceded by standard GN chemotherapy. Post-surgery, all patients were treated with adjuvant GN plus the CD40 agonist. In surgical specimens, neoadjuvant selicrelumab alone reduced fibrosis by approximately half compared to control untreated tumors. Further analysis of patient tumor and blood samples demonstrated a selicrelumab-dependent TME T cell enrichment and activation, higher density of mature DCs accompanied by decreases in M2-like TAMs within tumors, and increases in the inflammatory cytokines CXCL10 and CCL22 in the systemic circulation [142]. Failure to document previously observed increases in IL-12 were attributed to the later timepoint post-treatment at which samples were drawn. Correlation of these findings to what had been seen previously in mice was limited, re-emphasizing that pre-clinical models do not fully represent what happens in patients. Testing of selicrelumab is currently continuing as part of the MORPHEUS study, where it is being tested in combination with atezolizumab and GN (NCT02588443).

Sotigalimab (APX0005M) is a humanized rabbit IgG1 CD40 agonist antibody with very high potency that, unlike selicrelumab, does block the CD40L binding site. Combination of sotigalimab with GN, with or without nivolumab, was tested in patients with newly diagnosed metastatic pancreatic cancer in the PRINCE study [143]. Similar to the study of selicrelumab with GN, the CD40 agonist was administered 2 days after the first chemo dose in each cycle. In the PRINCE Phase 1b cohort, tolerability of the combination was demonstrated; the most common serious sotigalimab-related adverse event was pyrexia, seen in 20% of patients. Notably, 2 of 30 patients in the study died from complications attributed to the chemotherapy. A very encouraging 58% objective response rate was seen, prompting further study of the regimen. In the Phase 2 follow-up, participants were randomized to three arms: GN with nivolumab, GN with sotigalimab, or GN with both immunotherapy drugs. Only the chemo plus nivolumab cohort met the primary endpoint to improve upon the benchmark 1-year overall survival rate of 35% for GN alone. The study was not powered for comparison between arms, nor was a GN-only control arm included, so relative efficacy of the sotigalimab arms could not be evaluated; however, outcomes were numerically similar to the chemo plus nivolumab group. Longer overall survival in all arms was associated with a more diverse and immunocompetent T cell milieu pre-treatment. Similarly, the most discriminatory factor in identifying patients with longer survival in the sotigalimab plus chemo arm was higher in pre-treatment circulating DCs and B cells. Number of infiltrating CD8^+^ T cells was not associated with response in any group, unlike what has been seen with other types of solid tumors [144,145,146]. It was suggested based on these corelative findings that a biomarker-selected population might be required to see the benefit of this drug combination. [147].

Additional CD40 agonist drugs purported to have improved designs for anti-tumor activity continue to be developed. CDX-1140 is a fully human IgG2 CD40 agonist antibody that has been optimized to strongly stimulate the immune system while minimizing adverse immune-activation-related toxicity and was anticipated by its developers to permit higher doses/systemic exposures in the clinic compared to other CD40 agonist agents [148]. Consistent with this, 1.5 mg/kg was found to be the maximum tolerated dose in Phase 1 testing (NCT03329950), as compared to 0.2 mg/kg and 0.3 mg/kg for selicrelumab and sotigalimab (given on similar schedules), respectively [149]. In addition, CDX-1140 may exhibit enhanced DC stimulatory activity, since it does not block CD40L binding, thereby allowing DC stimulation with any native ligand that is present. As with other CD40 agonists, clinical evaluation in combination with anti-PD-1 is planned. However, a separate window of opportunity study in untreated resectable PDAC patients has begun enrollment to test the bioactivity of CDX-1104 alone as compared to combination with the recombinant Flt3 ligand CDX-301 (NCT04536077). Previous pre-clinical work has shown that DC dysregulation can be ameliorated, and conventional DC population numbers can be restored to normal levels with a combination of CD40 and the Flt3 ligand [150]. It will be interesting to see whether this study can demonstrate that a similar effect occurs in human patients.

SEA-CD40 is a humanized IgG1 CD40 agonist antibody which binds to APCs, induces enhanced crosslinking through FcγRIIIa, and can augment NK-cell binding to cancer cells [151]. Preliminary data from the ongoing Phase 1 study in first-line patients with PDAC (NCT02376699) suggest that the combination of SEA-CD40 with GN and pembrolizumab is tolerable [152], but it is too early to say whether this drug will have superior activity compared to sotigalimab.

Mitazalimab (ADC-1013; JNJ-64457107) is a fully human IgG1 which is FCγR-crosslinking-dependent. In pre-clinical testing, activity was demonstrated in combination with FOLFIRINOX in both chemo-sensitive and chemo-resistant mouse models [153]. The Phase ½ OPTIMIZE-1 study (NCT04888312) is currently enrolling adults with previously untreated metastatic PDAC to receive this combination. Preliminary information from the dose escalation phase suggests that the combination is safe [154].

An alternative platform for testing CD40 agonism is LOAd703, an oncovirus expressing trimerized CD40L and 4-1BB, another member of the TNF family [155]. LOAd703 is currently being evaluated in combination with standard chemotherapy in patients with advanced solid tumors including pancreatic cancer (NCT03225989). A subsequent Phase 2 trial designed to test LOAd703 with GN in patients with pancreatic cancer is ongoing (NCT02705196). A recently published interim analysis deemed the combination safe and tolerable, with toxicities comparable to other clinical CD40 agonists [141,142,156]. Most patients experienced treatment-emergent immune responses, including decrease in circulating MDSCs, increase in effector memory T cells, and rise in antigen-specific T cells. Interestingly, 6 out of 10 patients who received higher LOAd703 doses have had partial responses [157,158].

### 4.2. CXCR4-CXCL12 Axis

#### 4.2.1. Pre-Clinical Rationale

Chemokines are low molecular weight proteins belonging to the superfamily of cytokines that mediate immune cell adhesion, migration, and chemotaxis. They regulate immune-mediated inflammatory processes, as well as tissue injury reactions [159]. Apart from their physiologic role, chemokines are also involved in tumor progression by facilitating evasion from immune surveillance, inducing neoangiogenesis and infiltration of immunosuppressive cells, and potentiating distant metastasis formation [160].

Interaction between the chemokine CXCL12 and its receptor CXCR4 prominently features in communication between PDAC and its stroma. CXCL12 is secreted by CAFs [44] and coats PDAC tumor cells, since the malignant cells express higher levels of CXCR4 compared to normal pancreas tissue [161]. Studies performed on patient-derived tumor tissue showed that high CXCR4 expression correlates with the presence of more advanced and higher-grade tumors [162,163]. Additionally, higher CXCR4 expression was associated with worse overall survival [162].

In pre-clinical models of PDAC, higher expression of CXCL12 secreted by CAFs increased pancreatic cancer cell invasion [164] and promoted tumor growth by preventing circulating-T-lymphocyte (CTL) infiltration [165,166]. Cancer cell CXCR4 expression is mediated at least in part via Akt, HIF-1α, and NF-kB [167,168]. Coating of cancer cells with covalent CXCL12/keratin 19 heterodimers facilitated immunosuppression through this axis because these heterodimers excluded T cells from the immediate vicinity. They also conferred resistance to PD-1 checkpoint inhibitors [169]. Interestingly, CXCL12 secretion downregulates nociception in mouse models, suggesting this axis could be partially responsible for delays in PDAC diagnosis [170]. Increased CXCR4 expression also mediates resistance to the standard chemotherapy drug gemcitabine, and CXCR4 inhibition can mitigate this effect [168,171]. These data make the CXCR4-CXCL12 axis a tempting therapeutic target.

Multiple studies have demonstrated that CXCR4 inhibition can modify the PDAC TME. For instance, CXCR4 knockdown decreased the invasion potential of pancreatic cancer cells in vitro [161]. This was at least partially mediated via VEGF-independent inhibition of angiogenesis [172]. Treatment of fresh human PDA slice cultures with a combination of PD-1 and CXCR4 blockade had an anti-tumor effect with concomitant peripheral CD8^+^ T cell expansion [173]. In a PDAC mouse model, administration of a CXCR4 inhibitor AMD3100 (plerixafor) led to T cell accumulation among cancer cells with a synergistic tumoricidal effect when combined with a PD-L1 antagonist [44]. These results have prompted clinical assessment of CXCR4-CXCL12 axis modulation as an anti-PDAC strategy.

#### 4.2.2. Clinical Studies Targeting the CXCR4-CXCL12 Axis

A recent Phase 1 trial tested the immunological effects of a small molecule CXCR4 inhibitor AMD3100 (plerixafor) in patients with microsatellite stable (MSS) colorectal cancer and PDAC. Seven days of continuous AMD3100 infusion resulted in successful CXCR4 inhibition and decreases in surrogate markers of tumor burden such as circulating tumor DNA (ctDNA) and IL-8 [174,175]. Comparison of pre- and post-treatment biopsies showed a decrease in CAFs and increased tumor infiltration with T and NK effector cells, as well as an immune signature predictive of response to PD-(L)1 therapeutics in melanoma patients [176]. Given the transient duration of treatment, it was unsurprising that no radiographic responses were observed; however, this study provided important evidence that CXCR4 inhibition results in similar redistribution of lymphocytes in human PDAC patients as seen in pre-clinical mouse studies. Currently, plerixafor is being tested in metastatic pancreatic cancer patients in combination with the anti-PD-1 drug cemiplimab (NCT04177810). Obviously, the complexities of repeatedly administering a weeklong infusion, as required with AMD3100, present feasibility issues in clinical practice.

Motixafortide (BL-8040), a synthetic peptide with higher CXCR4 affinity and longer receptor occupancy, is administered by a more convenient subcutaneous dosing route [177]. In the Phase 2 COMBAT study, motixafortide with pembrolizumab was tested in previously treated patients with metastatic PDAC (NCT02826486). The disease-control rate was 34.5% with one partial response. Tumor biopsies showed increased CD8^+^ T cell tumor infiltration, decreases in MDSCs, and further decreases in circulating Tregs [178]. The second cohort of this trial enrolled patients with metastatic disease who had progressed on a first-line gemcitabine-based regimen. The standard chemotherapy regimen of 5-FU plus nal-iri from the NAPOLI trial was added to the dual immunotherapy. The treatment was well tolerated; toxicity was comparable or improved as compared to chemotherapy alone [32,179]. The overall response rate was 21% with a disease-control rate of 63.2% and at least one patient with a prolonged duration of response. Notably, no patients in the study had MSI disease. Efficacy benchmarks were numerically similar to those seen in the NAPOLI trial, although COMBAT accrued a poorer prognosis group of patients. These results are encouraging, but do not provide conclusive proof of immunotherapy efficacy given the lack of a control arm. A Phase 3 randomized study testing this regimen is anticipated. Currently, a study investigating motixafortide in combination with GN and cemiplimab is enrolling treatment-naïve patients with metastatic PDAC (NCT4543071).

Another approach towards targeting the CXCR4/CXCL12 axis is to inhibit CXCL12. Olaptesed pegol (NOX-A12) is a PEGylated mirror-image (L)-oligonucleotide, also called a Spiegelmer or L-RNA-aptamer, which binds CXCL12 and inhibits leukocyte chemotaxis at sub-nanomolar concentrations. It can also detach cell-surface-bound CXCL12 [180]. Similar to CXCR4 inhibitors, NOX-A12 increased tumor infiltration of T and NK cells and potentiated the activity of anti-PD-1 therapy in pre-clinical models [181]. The Phase 1/2 OPERA study (NCT03168139) enrolled patients with advanced, previously treated PDAC (and MSS metastatic colorectal cancer). The heavily pre-treated PDAC patients in the study had received a median of three prior therapies. While no radiologic responses were observed, two of nine PDAC patients had prolonged stable disease and remained in the study at least three times longer than for their most recent previous treatment. A trend towards increased effector immune cells in tumor biopsy tissue was observed. In addition, the combination had a favorable side effect profile comparable to single-agent pembrolizumab [182]. A new Phase 2 non-randomized study of NOX-A12 with pembrolizumab and either GN or 5-FU/nal-iri for second-line PDAC patients with MSS disease will shortly begin accrual (NCT04901741). The NOX-A12 dosing schedule has been intensified as compared to OPERA, given the favorable toxicity profile of the drug and concerns that simply priming with CXCL12 inhibition at treatment outset may be insufficient to maintain favorable immune cell profiles in the TME throughout the course of anti-PD-1 treatment.

### 4.3. Colony-Stimulating Factor Receptor (CSF-1R)

#### 4.3.1. Pre-Clinical Rationale

CSF-1R is a 165 kDa integral transmembrane glycoprotein with ligand-dependent tyrosine kinase activity [183]. CSF-1R is synthesized on membrane-bound polyribosomes, transported through the Golgi, undergoes glycolytic residue modification, and fuses with the cellular membrane via secretory vesicles [184]. The extracellular glycosylated ligand-binding domain contains five Ig-like domains, with domains 2 and 3 involved in binding. A variety of cell types express CSF-1R, including hematopoietic stem cells, monocytes, macrophages, osteoclasts, myeloid DCs, microglia, and Paneth cells [185]. CSF-1 or IL-34 bind to CSF-1R, causing phosphorylation of the intracellular tyrosine, which results in increased cell survival, proliferation, migration, and chemotaxis [186].

TAMs play a critical role in tumor development and the immunosuppressive phenotype of PDAC. PDAC secretes high levels of CSF-1, providing survival and proliferation signals to immunosuppressive CSF-1R+ macrophage infiltrates in the TME [187]. The presence of these immunosuppressive TAMs is thought to play a significant role in PDAC’s non-responsiveness to immunotherapy. In a murine PDAC model, a small molecule inhibitor of CSF-1R (ACD7507) prevented tyrosine phosphorylation of CSF-1R, resulting in depletion of tumor macrophages, increased survival, and decreased tumor burden. Fourteen days of ACD7507 treatment significantly decreased pro-tumor cytokines IL-6 and IL-10 in tumor samples [188]. Anti-CSF-1R therapies have also been shown to work well in combination with immune checkpoint inhibitors, causing sensitization to this class of agents and improved survival of murine PDAC models. In one combination study, the reduction in tumor growth correlated with an improved effector-to-regulatory T cell ratio [189]. In another, which also included the GVAX anti-tumor vaccine, the anti-CSF-1R antibody given with anti-PD-1 therapy increased the number of intratumoral PD-1+ CD8^+^ and PD-1+ CD4^+^ T cells and increased their expression of IFNγ, a cytokine known to stimulate NK cells and neutrophils [190]. Nanomicelle formulations containing the PI3Kγ inhibitor with CSF-1R-siRNA have also been used to treat mice with PDAC. Compared to the control groups, treatment with this combination resulted in statistically significant increases in M1 macrophages, decreases in the M2 macrophage population, increased numbers of CD8^+^ and CD4^+^ T cells within the TME, and a statistically significant decrease in the concentration of IL-10 [191]. In pre-clinical models, CSF-1R inhibition consistently results in a less immunosuppressive PDAC TME.

#### 4.3.2. Clinical Data with CSF-1R Inhibitors

Pexidartinib (PLX3397) is a multi-kinase inhibitor targeting CSF-1R that is FDA approved for the treatment of tenosynovial giant cell tumor, a rare CSF1-driven neoplasm [192]. Common side effects of pexidartinib include hepatic transaminase abnormalities which can (rarely) be associated with fatal liver injury, hypercholesterolemia, elevated lactate dehydrogenase, and hair color changes. In the Phase 1 MEDIPLEX study, pexidartinib was tested in combination with durvalumab in patients with PDAC and CRC. Toxicities were tolerable and similar to those of the single agents. The clinical benefit rate at 2 months for those on the dose escalation cohort was 21%, as 4 of 19 patients had stable disease. Although enrollment of the dose expansion cohort was completed in January 2019, the results have not yet been reported [193].

AMG 820 is a fully human IgG2 mAb targeting CSF-1R that blocks binding of CSF1 and IL-34 ligands. First-in-human testing (NCT01444404) identified a dose-limiting toxicity of irreversible hearing loss in one participant [194]. More common toxicities included reversible periorbital edema (which had also been observed in animal toxicologic studies and is of unknown etiology) and elevated liver transaminases. The latter likely occurs secondary to disrupted liver enzyme homeostasis due to on-target depletion of Kupffer cells; no evidence of true hepatotoxicity was observed. Due to evidence of bioactivity but lack of single-agent anti-tumor activity, AMG 820 was advanced into a Phase 1/2 combination study with pembrolizumab (NCT02713529) without completing planned dose expansion cohorts. Eligible patients included those with advanced pancreatic cancer refractory to standard-of-care treatments. In the PDAC cohort, 10/26 participants had best response of immune-related stable disease, with 2 having numerical decreases in tumor diameter. This was insufficient to meet predefined threshold criteria for efficacy, although the limited number of paired tumor biopsies examined did demonstrate the expected bioactivity, including a reduction in CSF1-dependent CD16-expressing monocytes, and increased CD4^+^ and CD8^+^ T cell numbers [195].

One additional pilot study is investigating the bioactivity of CSF-1R inhibitor IMC-CS4 (LY3022855) given in combination with pembrolizumab and the cyclophosphamide/GVAX anti-pancreatic cancer vaccine to patients with borderline resectable pancreatic cancer (NCT03153410). The co-primary objectives of this study are safety and a biologic endpoint of change in CD8^+^ T cell density in the primary tumor with treatment. The study is not designed to assess for improvements in clinical outcome; however, any radiologic responses seen in the treatment group would be notable.

Overall, results of the completed studies utilizing CSF-1R inhibitors suggest that combination of these biologically active drugs with anti-PD-1 therapy alone is insufficient to produce significant anti-tumor activity in advanced PDAC.

### 4.4. CD11b

CD11b is an integrin heterodimer formed with CD18 and is expressed on a variety of cell types including MDSCs, TAMs, and DCs. Its primary role in oncogenesis is to improve myeloid cell adhesion to vasculature, tissue recruitment under inflammatory conditions, and survival [196]. Since CD11b+ MDSC populations are increased in PDAC patients and are thought to suppress the anti-tumor immune response, CD11b is considered a viable therapeutic target [197].

ADH-503 (GB1275) is a small molecule that binds to the allosteric pocket of CD11b and stabilizes it in an active state. This augments adhesion of myeloid cell CD11b to receptors on vascular endothelium, impairing myeloid cell migration into tissues. The forced CD11b activation state also shifts TAM polarization to a more anti-tumor phenotype. Neither of these activities require target saturating concentrations of drugs, an important boon over a true pharmacologic inhibitor [196]. In a murine PDAC model, ADH-503 reduced myeloid cell recruitment to tumor tissues and reprogrammed the remaining macrophages to have a more M1-like phenotype, resulting in increased effector T cell frequency and closer proximity of these lymphocytes to the cancer cells. Moreover, single agent ADH-503 improved survival in tumor-bearing mice and sensitized PDAC tumors to anti-PD-1/PD-L1 immunotherapy [198]. In patients (NCT04060342), ADH-503 was well tolerated, with the most common side effects being photosensitivity, dysesthesia, and pruritus. No unexpected toxicities were observed with addition of pembrolizumab; however, preliminary reports have indicated no clinical responses in pancreatic cancer patients [199]. Final results of this Phase 1/2 study are still awaited.

### 4.5. STING (Stimulator of Interferon Response cGAMP Interactor) Pathway

STING is a transmembrane endoplasmic reticulum protein that is activated through binding of cytosolic DNA and cyclic dinucleotides (CDNs) [200]. Activated STING induces expression of type I IFN not only in response to bacterial and viral pathogens [201,202], but also to CDNs produced by cancer cells [203]. DCs detect CDNs released by cancer cells, and subsequently prime CTLs against the tumor [204,205]. Pre-clinical models have demonstrated that STING1-deficient mice cannot mount an efficient T cell response against syngeneic gliomas [206] and melanomas [207]. STING-mediated signaling is necessary for spontaneous T cell activation by cancer [207].

In mice bearing PDAC tumors, combination of a neoantigen-targeted vaccine with a STING agonist adjuvant led to transient tumor regressions. When immune checkpoint modulators were added to this cocktail, more durable tumor regressions were observed, survival increased, and tumor rejection was elicited on rechallenge [208]. In another syngeneic murine pancreas cancer model, treatment using a STING agonist (DMXAA) with gemcitabine increased animal survival. Complimentary immune system activation was also observed, including production of inflammatory cytokines, increases in maturation markers on DCs, and augmentation of the functional tumor-infiltrating CTL population [209]. The same group later reported that single-agent administration of another novel STING agonist (ADU-S100) decreased tumor burden and activated the murine immune system by increasing CTL tumor infiltration and decreasing TAMs and Tregs in a CXCR3-dependent fashion [210]. Other studies using intratumoral administration of several CDN STING agonists in a KPC-derived orthotopic murine model found that high-potency CDNs diminish proliferation of MDSCs and TAMs through downregulation of Myc signaling and prolong mouse survival independent of chemotherapy through potentiation of checkpoint inhibitors [211]. These studies form the basis for clinical studies of STING agonist therapeutics.

Clinical testing of STING agonists is in its infancy. A Phase 1 study of a CDN STING agonist, MK-1454, alone or in combination with pembrolizumab, established the safety of the drug (NCT03010176). The most common adverse events were pyrexia, fatigue, nausea, and pruritus [212]. A second STING agonist, TAK-500, has recently begun clinical testing alone or in combination with pembrolizumab in patients with advanced cancers, including PDAC (NCT05070247).

### 4.6. CCR2

PDAC highly expresses the CCL2 chemokine which leads to mobilization of CCR2+ monocytes from the bone marrow to the tumor. Increased monocytosis and mobilization from the marrow are associated with worse prognosis [213]. Once in the tumor, the mobilized monocytes transform into TAMs which exhibit immunosuppressive properties [214]. Higher TAM densities correlate with poor prognosis [215]. Targeting the CCL2/CCR2 axis in PDAC would be anticipated to reverse this accumulation of TAMs in the TME.

A CCR2 inhibitor PF-04136309 was tested in a phase 1b trial in combination with FOLFIRINOX in patients with borderline-resectable and advanced PDAC (NCT01413022). A tolerable safety profile was demonstrated. While initial correlative studies had shown a reduction in TAMs and influx of tumor-infiltrating lymphocytes [216], subsequent analyses demonstrated that CCR2 inhibition resulted in compensatory increases in CXCR2+ TANs, putting into question the rationale of targeting a single myeloid subset [217]. Another trial combined PF-04136309 with GN in patients with metastatic PDAC (NCT02732938). In this trial, an unexpectedly high rate (24%) of pulmonary toxicity was observed, and the efficacy signal was similar to previously reported benchmarks for the chemotherapy alone [218]. There are currently no active Phase 2 trials testing this agent in PDAC patients.

Another CCR2 antagonist, CCX872-B, was tested in combination with FOLFIRINOX in patients with advanced pancreatic cancer (NCT02345408). High receptor occupancy was observed at tolerable doses [219]. Interim analysis showed overall survival of 29% at 18 months with no safety issues. This benchmark was noted to compare favorably to that seen in the Phase 3 study which established FOLFIRINOX as a standard of care [1]. Lower peripheral blood monocyte counts at baseline were associated with improved overall survival [220]. Although these preliminary results were reported in abstract form in 2018, no final publication has yet appeared in the literature and no follow-up studies are registered.

BMS-813160 is a small molecular CCR2-CCR5 dual antagonist that was tested as a monotherapy or in combination with chemotherapy or nivolumab in patients with advanced pancreatic or colorectal cancer (NCT03184870). The study began accrual in 2017 but no results have yet been reported [221].

### 4.7. CD73/A2A Adenosine Receptor

CD73, also known as ecto-5′-nucleotidase (NT5E), is a cell surface enzyme that catalyzes the conversion of AMP (adenosine monophosphate) to adenosine. Because free adenosine triggers inhibition of T cell receptor activation through lymphocyte adenosine receptors, increased CD73 activity is considered immunosuppressive. Adenosine produced by cancer cells, CAFs, and myeloid cells contributes to the immunosuppressive nature of the TME. Mice with genetic knockout of the A2A adenosine receptor have increased ability to reject tumors [222], a more robust population of tumor antigen-specific CD8^+^ T cells in draining lymph nodes, and enhanced response to anti PD-1 therapy. Improved activity of anti-PD-1 therapy was also seen with pharmacologic inhibition of A2A adenosine receptor [223]. Numerous anti-CD73 therapeutics and other adenosine receptor inhibitors have since been developed (reviewed in [224]).

Oleclumab (MEDI9447) is an IgG1λ mAb that binds CD73 and causes its endocytosis. Treatment of tumor-bearing mice with oleclumab slowed tumor growth and enhanced CD8^+^ and CD4^+^ T cell infiltration in colon cancer models. Combination with anti-PD-1 led to tumor rejection in 60% of animals [225]. Clinical testing of oleclumab with or without durvalumab was initiated in patient populations known to be unresponsive to anti-PD-(L)1 therapies: advanced pancreatic cancer, MSS colorectal cancer, and EGFR mutant non-small cell lung cancer (NCT02503774). Both single-agent and combination regimens were well tolerated. Two of seventy-three PDAC patients treated in the study achieved partial responses with durations of response of 22+ and 28+ months [226]. The full report of study outcomes has not yet been published. Similarly, no information is available for the follow-up Phase 1/2 study in advanced PDAC testing of the oleclumab/durvalumab combination with standard chemotherapy (NCT0361156). However, a planned subsequent study (NCT04262375) was listed as withdrawn due to insufficient activity of the oleclumab/durvalumab doublet.

Phase 1 clinical trials (NCT03207867, NCT03549000) testing the A2A adenosine receptor inhibitor taminadenant (NIR178) with anti-PD-1 spartalizumab and/or anti-CD73 fully human antibody NZV930 were accruing patients with pancreatic cancer and many other tumor types, but no data have yet been reported.

Quemliclustat (AB680) is a selective, reversible, and competitive small molecule inhibitor of CD73 with picomolar affinity. This first-in-class drug is likely to have improved tumor penetration compared to monoclonals due to its smaller size. Despite this, it retains a lengthy half-life suitable for parenteral dosing. In mouse models, treatment with quemliclustat improved T cell function, induced CD8^+^ T cell infiltration into tumors, and reduced animal tumor burden in melanoma models [227]. In vivo evaluation of the drug in pancreas cancer models has not been published. In the ARC-8 study (NCT04104672), the safety and tolerability of quemliclustat in combination with standard GN and the anti-PD-1 drug zimberelimab was evaluated in patients with treatment-naïve metastatic PDAC. The safety profile has thus far resembled that of the single agents, with no additional quemliclustat-related toxicities. Multiple partial responses have been observed with prolonged duration of response [228]. Per company press releases, a randomized control arm is expected to be added to this study to make a clearer assessment of the relative contribution of quemliclustat to the observed clinical activity.

## 5. Targeting the Stroma

Drugs that directly target non-immune cells in the stroma have also moved into the PDAC space. The first such studies, using small molecule inhibitors of Hh signaling to ablate stromal fibroblasts, built on the idea that stromal fibroblasts were purely tumor supportive and that the dense ECM that they constructed served primarily to limit chemotherapy delivery and effectiveness [229]. Unfavorable results in clinical testing of Hh inhibitors vismodegib and IPI-926 prompted re-examination of the role these cells play in pancreatic cancer and led to our current understanding that tumor fibroblasts have both tumor-restraining and tumor-promoting effects [47,48,230]. Subsequently, therapeutic targeting of this compartment has relied on strategies which modulate precise cell populations, signaling molecules produced by fibroblasts, or ECM components within the stroma (Table 2).

### 5.1. FAK Inhibitors

FAK1 and FAK2 are nonreceptor tyrosine kinases with varying roles. FAK1 helps to induce pro-inflammatory pathways that lead to Treg activation and CD8^+^ T cell inhibition in murine cancer models [231]. FAK1 has also been implicated in development of pathologic fibrosis [232], including maintenance of the desmoplastic stroma and the TAM population in PDAC TME [233]. In KPC mice, FAK inhibition reduced tumor fibrosis and immunosuppressive cell populations, rendering tumors more sensitive to chemotherapy and PD-1 blockade [234].

FAK inhibition with defactinib is being tested in combination clinical trials. These include accruing studies examining defactinib combined with PD-1 blockade (NCT02546531) in the neoadjuvant setting, or with stereotactic body radiation (NCT04331041) in patients with locally advanced PDAC. In addition, a Phase 1 study of defactinib plus pembrolizumab and gemcitabine in patients with advanced tumors has been completed. This regimen was well tolerated and a recommended Phase 2 dose of defactinib was established. Tumor control was observed: 54% of evaluable patients had stable disease (NCT02546531). Testing in a PDAC-only expansion cohort is ongoing [235].

A second small molecule FAK inhibitor, GSK2256098, has also been tested in advanced treatment refractory PDAC. The MOBILITY-002 trial examined combination of GSK2256098 with the MEK1/2 inhibitor trametinib (NCT02428270). While the combination was safe, no clinical activity was observed [236].

### 5.2. IL-6

IL-6 participates in PDAC genesis and progression [237] and high levels of IL-6 in PDAC are associated with worse overall survival [238]. A recent study demonstrated that PDAC stroma is a major source of IL-6 in PDAC patient samples. Blockade of IL-6 and PD-1 produced anti-tumor activity in several mouse models. This was driven by intratumoral infiltration of CD8^+^ T cells and accompanied by reduction in αSMA+ cells [239].

Siltuximab is a chimeric IgG1 anti-IL-6 mAb that was tested as a single agent in KRAS-mutated solid tumors. No clinical activity was observed for the single agent [240]. However, an ongoing Phase 1/2 study is re-evaluating siltuximab in combination with the PD-1 inhibitor spartalizumab (NCT04191421).

The Phase 2 PACTO study is testing the efficacy of adding tocilizumab, a humanized IgG1 anti-IL-6 mAb, to standard-of-care GN in treatment-naïve patients with advanced PDAC (NCT02767557). While the study was initiated in 2016 and is no longer accruing, no results have been posted. Conversely, the Phase 2 TRIPPLE-R study (NCT04258150) of tocilizumab, SBRT, and dual-checkpoint inhibitor blockade (ipilumimab + nivolumab) as second-line treatment for pancreas cancer patients with advanced disease was terminated for not meeting the primary endpoint. The results of this negative study have not yet been reported. Despite these negative results with tocilizumab, the Phase 1/2 MORPHEUS study (NCT03193190) includes an arm testing the combination of tocilizumab, atezolizumab, and GN.

### 5.3. Vitamin D

Due to the presence of exocrine pancreas dysfunction, patients with PDAC suffer from dietary deficiencies, including insufficiency of fat-soluble vitamins such as vitamin D [241]. CAFs unexpectedly express high levels of vitamin D receptor (VDR), now known to be a major transcriptional regulator of CAF activation, but have decreased expression of lipid storage and metabolism genes, with consequent loss of the lipid droplet present in more quiescent cells [242]. Treatment with calcipotriol, a vitamin D analog, reverts CAFs back to a quiescent state, reduces tumor-associated fibrosis, and potentiates gemcitabine efficacy in KPC mice, at least partially by increasing local gemcitabine concentration [242]. Interestingly, while calcipotriol decreases CAF proliferation, migration, αSMA expression, and secretion of pro-tumorigenic factors such as IL-6, a more recent study showed that it also promotes PD-L1 upregulation, accompanied by reduction of T cell effector function in patient-derived 2D and 3D cell culture models. This suggests that while vitamin D analogs remodel the PDAC TME to a more immune favorable environment, they could also compromise tumor immune surveillance [243]. Nevertheless, vitamin D insufficiency correlates with worse prognosis in PDAC patients [244], and addition of vitamin D analogs is being pursued in multiple clinical trials.

The biologic effect of adding paricalcitol to standard GN was assessed in a completed pilot neoadjuvant window-of-opportunity study (NCT02030860) which has yet to report results. Results from a second neoadjuvant window-of-opportunity study combining paricalcitol plus pembrolizumab with or without GN are also unpublished (NCT02930902). Subsequent to initiation of these trials, Phase 1 and 2 combination studies of paricalcitol treatment in the advanced disease setting began enrollment. Results of a Phase 2 trial testing nivolumab, nab-paclitaxel, paricalcitol, cisplatin, and gemcitabine have been presented in abstract form. A total of 32 patients were evaluable with an impressive response rate of 84%, a benchmark similar to that seen for the triplet chemotherapy regimen alone [245]. Most common drug-related grade 3–4 AEs were thrombocytopenia (76%) without serious bleeding events, anemia (37%), and chemotherapy-induced neutropenia (11%). Full analysis, including reporting of exploratory endpoints, is pending [246]. Other studies include an ongoing Phase 2 trial of paricalcitol combined with GN (NCT03520790) for patients with previously untreated metastatic PDAC, and a separate study (NCT03331562) testing pembrolizumab maintenance with placebo or paricalcitol following development of best clinical response on first-line chemotherapy in patients with metastatic disease. The latter has been recently completed. Preliminary results posted on clinicaltrials.gov suggest that addition of paricalcitol did not prove beneficial. Publication of full results in a peer-reviewed format is still awaited.

### 5.4. All-Trans Retinoic Acid (ATRA)

CAFs can be restored to a quiescent phenotype through exposure to fat-soluble vitamins, such as vitamin A. Administration of ATRA to KPC mice induces CAF quiescence, with desmoplastic stroma collapse, tumor growth inhibition [247], as well as CD8^+^ T cell infiltration [248]. Given these results, addition of ATRA to other therapeutic modalities is being pursued in PDAC.

The Phase 1 trial STAR_PAC study (NCT03307148) combined ATRA with GN in patients with advanced PDAC naïve to chemotherapy in the advanced disease setting. A favorable safety profile was observed even with full dose chemotherapy; in fact, the investigators reported lower neurotoxicity than typically seen with nab-paclitaxel [249]. The follow-up STARPAC2 randomized Phase 2 study comparing GN with ATRA to the chemo alone is currently enrolling (NCT04241276). This study is specifically for patients with untreated locally advanced disease that is proven metastasis-free by exploratory laparotomy.

### 5.5. Vascular Endothelial Growth Factor (VEGF)

VEGF is a critical promoter of angiogenesis in both normal physiology and in tumors. Secretion of VEGF is upregulated by activation of Hypoxia-Inducible Factor-1α under hypoxic conditions (reviewed in [250]), like those present in PDAC [251]. VEGF is expressed in PDAC and is correlated with higher microvascular density and poorer prognosis [252,253]. However, blockade of VEGF signaling with biologics such as bevacizumab [254,255], or ziv-aflibercept [256], or with receptor tyrosine kinase inhibitors such as sunitinib [257], axitinib [258], or regorafenib [259] was tested extensively in PDAC patients in combination with standard chemotherapy or erlotinib and offered no clinical benefit. Use of anti-VEGF agents in PDAC was largely abandoned after these failures.

Resurgent interest in VEGF inhibition has now emerged when given in combination with anti-PD-(L)1 therapeutics [260]. In addition to stimulating angiogenesis, VEGF is also a potent suppressor of anti-tumor immunity. VEGF signaling leads to accumulation of inhibitory immune cell populations within tumors, promotes T cell exhaustion [261], impairs maturation of DCs [262], and produces abnormal, tortuous, and leaky blood vessels which suppress infiltration by effector leukocytes in part by reducing expression of endothelial cell surface leukocyte adhesion molecules (reviewed in [263]). Combination of anti-angiogenic agents with immune checkpoint inhibitor therapy has produced favorable clinical outcomes for patients with hepatocellular carcinoma [264], renal cell carcinoma [265], non-small cell lung cancer [266], and endometrial cancer [267]. In addition, there is a case report of a heavily pre-treated PDAC patient achieving a complete response to the combination of pembrolizumab and lenvatinib [268].

Recently, several clinical trials testing combinations of anti-PD(L)1 drugs with anti-angiogenic agents have opened. LEAP-005 is a Phase 2 study testing lenvatinib with pembrolizumab in patients with multiple tumor types who have advanced disease that has progressed on prior treatment (NCT03797326). Preliminary data from all initial cohorts of the LEAP-005 study have been reported in abstract form. Notably, in the MSS colorectal cohort an overall response rate of 22% was seen with duration of response not reached [269]. This is encouraging given that clinical activity of single-agent and dual immune checkpoint inhibitor therapy is not usually seen in this population. The field continues to await data on the PDAC cohort which began accrual in March 2021 [270]. The lenvatinib plus pembrolizumab combination is also being tested in PDAC for the maintenance setting in patients who have reached their best response with first-line chemotherapy (NCT04887805). In addition, combination of lenvatinib with durvalumab and nab-paclitaxel (NCT05327582) or with oncolytic virus H-101 and anti-PD-1 inhibitor tislelizumab (NCT05303090) began enrollment in 2022.

The combination of bevacizumab with atezolizumab and standard GN chemotherapy is being tested on one arm of the MORPHEUS-PDAC clinical trial (NCT03193190) [271]. No data are yet available for the cohort containing anti-angiogenic agent.

### 5.6. Integrins

Integrins are heterodimeric cell surface receptors that play critical roles in both cell adhesion to ECM and bidirectional cell signaling [272]. Some play important roles in angiogenesis, including tumor angiogenesis.

Integrin α_5_β_1_ interacts with fibronectin in ECM to provide important survival signals that activate endothelial cells and is upregulated in many tumor types (reviewed in [273]). Volociximab is a chimeric mAb (IgG4) that binds α_5_β_1_ integrin and blocks its association with fibronectin. Pre-clinically, volociximab prevented neovascularization at nanomolar concentrations [274] and inhibited angiogenesis in human tumor xenografts [275]. In clinical testing, volociximab had an acceptable safety profile but demonstrated insufficient clinical activity to warrant further testing even when given in combination with gemcitabine to pancreatic cancer patients [276,277].

Integrin α_V_β_3_ is expressed on angiogenic endothelial cells, activated macrophages, and collagen-secreting myofibroblasts [278,279,280,281]. It has been detected at high levels in invasive cancers [282], including PDAC, and most particularly in PDAC lymph node metastases [283]. Recent data have demonstrated that activated fibroblasts, such as CAFs, express high levels of integrin α_V_β_3_, while quiescent fibroblasts do not [284,285]. ProAgio is a rationally developed protein cytotoxin designed to target integrin α_V_β_3_ at a novel site. Unlike prior anti-integrin α_V_β_3_ therapeutics, ProAgio does not just block integrin signaling. Instead, binding of ProAgio induces apoptosis of integrin-α_V_β_3_-expressing cells by recruiting and activating caspase 8 to the cytoplasmic domain of β_3_ through a novel mechanism [284]. Pre-clinical studies have shown activity of ProAgio in mouse models of multiple cancer types including PDAC, and that combination with gemcitabine or immunotherapy enhances single agent activity [286,287]. We are currently testing ProAgio in a Phase 1 dose-escalation study including an expansion phase specific for pancreatic cancer patients (NCT05085548).

### 5.7. Hyaluronan

High fluid pressure within the PDAC TME causes collapse of functional blood vessels and impedes delivery of even small molecule therapeutics to cancer cells. Hyaluronan is a major component of the extracellular matrix and forms a hydrated gel that increases interstitial fluid pressure within tumors, leading to vascular collapse. Enzymatic degradation of hyaluronan in the KPC spontaneous autochthonous model of pancreatic cancer reversed this process [33]. PEGPH20, a clinically formulated PEGylated human recombinant PH20 hyaluronidase, increased gemcitabine delivery to these mouse PDAC tumors and inhibited tumor growth [288]. Phase 1 testing of PEGPH20 with gemcitabine demonstrated the safety of the combination and uncovered an advantage in clinical outcomes for treated patients with hyaluronan high tumors [289]. This led to the Phase 2 HALO 202 study combining PEGPH20 with GN, where favorable results were seen in patients with high stromal density, a pre-planned subgroup analysis [290]. Subsequently, the Phase 3 HALO 109-301 trial, which specifically accrued patients with hyaluronic acid high disease, was initiated. Despite restricting enrollment to the subgroup most likely to benefit, the novel combination did not lead to improvement in patient survival compared to GN alone [291]. Similarly, activity of PEGPH20 combined with atezolizumab proved to have insufficient activity for further study, with an overall response rate of only 6.1% in the Phase 1b/2 MORPHEUS study [292]. Surprisingly, combination of FOLFIRINOX with PEGPH20 was found to be detrimental in a randomized Phase 2 study, mainly due to a high incidence of gastrointestinal and thromboembolic adverse events [293]. At this point, no further studies of PEGPH20 in PDAC are being pursued.

### 5.8. Losartan

Collagen is required for increased hyaluronan to amplify interstitial fluid pressure and vascular collapse in PDAC [46]. Losartan is an angiotensin II receptor blocker (ARB) that is widely prescribed as an anti-hypertensive. However, in addition to its blood pressure lowering effects, losartan also reduces fibrosis in hypertensive kidneys by preventing injury-stimulated expression of TGF-β [294]. Testing of low-dose losartan in multiple mouse tumor models demonstrated that the drug could reduce deposition of TME collagen I [295] and hyaluronan, reduce CAF density, lower tumor solid stress, decompress tumor blood vessels, enhance drug delivery, and augment the anti-tumor effect of chemotherapy primarily through blockade of an angiotensin II receptor [46]. Interestingly, increases in mouse tumor vessel size and fractional blood volume induced by losartan treatment could be non-invasively assessed by MRI using magnetic iron oxide particles [296]. At the same time, losartan blockade of an angiotensin I receptor is reported to cause anti-angiogenic effects by reducing expression of VEGF [297,298]. Repurposing of losartan for anti-cancer indications was anticipated to produce no safety concerns, except that many PDAC patients may not have sufficient excesses in blood pressure to feasibly tolerate an anti-hypertensive agent.

The effect of losartan treatment in combination with FOLFIRINOX was initially tested in a Phase 1 study accruing ACE- and ARB-inhibitor-naïve PDAC patients with locally advanced disease (NCT01821729). Patients received neoadjuvant FOLFIRINOX with losartan followed by radiation. Tolerability of adding at least 25 mg of daily losartan was tested in a 1-week lead-in period that commenced simultaneous to the first FOLFIRINOX administration. If well tolerated, the losartan dose was advanced to 50 mg daily. Only 3 of 49 patients in the study experienced hypotension and all were able to continue study treatment. Forty-five patients proceeded to radiation, and thirty-four had sufficient tumor response to undergo resection. The primary endpoint of the study, to increase the R0 resection rate to 25% or greater from a historical benchmark of 10% or less, was easily met, with 30 of the initial 49 patients (61%) achieving this landmark [299]. This exciting single-arm study was not designed to detect differences in overall survival; however, a randomized Phase 2 follow-up study is currently enrolling a similar population of patients to receive losartan and chemoradiation with or without nivolumab (NCT03563248). In addition, the Phase 1 SHAPER study is examining the safety of giving losartan with hypofractionated radiation in a similar patient population (NCT04106856).

Several other studies examining losartan have recently begun enrollment. The Phase 2 NeoOPTIMIZE study (NCT4539808) is accruing patients with resectable, borderline resectable, or locally advanced disease to receive a regimen of mFOLFIRINOX with losartan for up to four cycles followed by continuation of losartan through chemoradiation. Those who progress on or are intolerant to FOLFIRINOX will be switched to GN/losartan for two cycles before starting chemoradiation. The primary endpoint is once again the R0 resection rate. Another new Phase 1 feasibility study is combining losartan with two other purported stroma-modifying drugs, hydroxychloroquine and paricalcitol, in a window-of-opportunity study for resectable PDAC patients (NCT05365893). The novel regimen will be given following neoadjuvant mFOLFIRINOX and radiation, and its effect on the TME will be assessed in surgical specimens. Pre-clinical studies of this regimen have not been reported. Losartan treatment is also being testing in the therapy-naïve metastatic PDAC population. A new four-arm Phase 2 study is enrolling these patients to FOLFIRINOX, FOLFIRINOX + losartan, FOLFRINOX + GSK-3β inhibitor elraglusib (9-ING-41), or chemotherapy with both novel agents (NCT05077800). There are no published pre-clinical data examining the combination of FOLFIRINOX or losartan with elraglusib.

## 6. Summary and Perspective

Many novel agents targeting key pathways in the PDAC TME are currently undergoing testing. The field is anxiously awaiting results from early studies of exciting new agents, and for definitive follow-up studies from those with promising reports in early-stage trials. A few possible success stories have been identified. Studies with CD40 agonist drugs have suggested that these agents could have modest clinical benefit in PDAC patients with advanced disease using current combination strategies. Movement of these agents to the resectable disease setting may produce greater benefit. Addition of losartan to neoadjuvant chemotherapy has clearly produced exciting results that may translate into more cures for patients with localized disease. The addition of effective new agents with tolerable side effects to the current anti-PDAC arsenal would be a boon to patients and providers.

In our age of precision medicine, a new emphasis has been placed on understanding which sub-populations of patients on trial are most likely to benefit from a targeted agent. In the cases of erlotinib for EGFR mutant lung cancer, trastuzumab for HER-2 amplified breast cancer, olaparib for patients bearing the BRCA mutation, or anti-PD-1 therapy for MSI-H/dMMR tumors, the correlation between biomarker and clinical response is difficult to miss. Clearcut determinants of what defines the rare responder to oleclumab/durvalumab or motixafortide/pembrolizumab are not so obvious at this point. Extensive correlative analyses to characterize tumors with improved responses was performed on the recently reported PRINCE study testing the CD40 agonist sotigalimab with durvalumab and GN [147]. Most of the candidate circulating biomarkers identified were indeed predictive rather than prognostic, as they were associated with only one of the three study treatment regimens. However, it is still not straightforward how this information could be applied in designing the next trial. Presently, it is not clear whether the anticipated magnitude of differential benefit between biomarker-positive and biomarker-negative persons would be clinically meaningful enough to warrant biomarker use when selecting eligible candidates for a future study. Nor is it clear how an appropriate cut point could be established, though admittedly, cut points for most eligibility-determining integral biomarkers are largely arbitrary. Notably, putative biomarkers for immune- and TME-modifying therapeutics have not been a sure bet. Tumor PD-L1 expression is not predictive of response to anti-PD-(L)1 therapies in some tumor types, but completely concordant in others. The negative Phase 3 HALO 109-301 study utilized a mechanistically relevant biomarker identified in a pre-planned subgroup analysis from the proceeding Phase 2 study to narrow down the patient population accrued, yet this did not expose the expected treatment benefit of PEGPH20 [300]. Given the complexity of immune and TME interactions, one might hypothesize that co-evaluation of multiple biomarkers may be required to identify patients with the greatest chance of response.

Many of the well-designed agents discussed in this article have not panned out clinically. Accompanying correlative studies have, in most cases, suggested failure is not due to lack of bioactivity. PEGPH20 and inhibitors of CSF1R, CD11b, and CCR2 all demonstrated their expected biological effects on the human patient immune and stromal compartments, yet clinical outcomes from these studies are far inferior to the strong anti-tumor effect these drugs provoked in mouse models. In addition, combinations of anti-PD-(L)1 therapy with CXCR4/CXCL12 or CD73/A2AR inhibitors appear to produce infrequent, isolated patient responses rather than the reliable anti-tumor activity suggested in pre-clinical testing. For many (if not all) of these agents, pre-clinical testing was performed in what are considered gold standard models of PDAC using well-designed experiments with careful controls, yet these experiments failed to prospectively anticipate an efficacy failure mechanism. Unfortunately, this is not a problem unique to the PDAC field—half of all experimental drugs fail due to inadequate efficacy [301]—but it does frequently feel as though this tumor type presents special challenges. The field has continued to see little correlation between clinical benefit in PDAC patients and anti-tumor activity in pre-clinical models, including resource-intensive, genetically engineered autochthonous mouse models that appear to closely resemble the human disease histologically and genetically [302]. Examining therapeutics designed to act on the TME requires faithful representation of the complex interactions that occur between multiple cell types which may not exhibit exactly concordant biology in mice versus humans. Human-derived models such as patient-derived xenografts and organoids may be more successful at predicting responses to chemotherapy and inhibitors of oncogenic drivers; however, these models have serious limitations when assessing TME-modifying agents [303]. For instance, the requirement for an immune-competent system mostly necessitates the use of non-human tumors and testing of an anti-murine version of the clinical mAb, rather than the actual clinical mAb, since most anti-human mAb do not cross-react with their mouse orthologues. In addition, implanted tumor models may not have sufficient time to develop a mature TME before experimental interventions are initiated. Moreover, no matter how long researchers wait before initiating treatment, tumors in mice will never grow as large as those that clinicians are seeing on their patients’ CT scans. These limitations have been noted [302], but alternatives are not necessarily available or feasible.

With chemotherapies and inhibitors of oncogenic drivers, demonstration of strong single-agent activity forms the basis for further testing and development. By contrast, all TME-modulating therapeutics discussed here have low expectation of producing clinical benefit as single agents. It is anticipated that a combinatorial approach utilizing multiple targeted agents within our growing therapeutic toolkit will be required to successfully overcome therapeutic resistance in most PDAC patients. The ever-expanding list of clinical TME-modifying agents available for combination presents researchers with a welcome challenge of riches, as limitations on both resources and patient population prevent empiric evaluation of all possible permutations in the clinical setting. Moreover, varying the dosing schedule, treatment duration, and/or drug sequencing can diametrically alter anti-tumor effects, making it too easy to fail even with a well-considered cocktail of complimentary agents [304]. For instance, the possibly reduced benefit/lack of increased benefit of the dual immunotherapy arm in the PRINCE study as compared to the single immunotherapy arms was attributed to excessive immune stimulation [147]. The requirement for rational combination increases the complexity of meaningful assessment.

Novel clinical trial designs are necessary to efficiently co-examine and compare the effects of multiple TME-modulating agents. Some researchers have already begun an iterative process of testing a bioactive agent, identifying the compensatory pathways that block clinical anti-tumor efficacy, then bringing forward a next clinical trial which adds a new drug expected to counteract tumor resistance. By contrast, the MORPHEUS study packs evaluation of 2 active comparator arms (GN or mFOLFOX6) and 10 experimental arms testing various atezolizumab combinations into a single study that can potentially add even more experimental arms. There are multiple advantages to this design that offset the unwieldiness. First, the presence of active comparator arms allows for efficient present-time comparison against the standard of care. While it is frequently infeasible to include a standard-of-care control arm on Phase 2 studies because patients are unwilling to continue the study if randomized to drugs they could receive off-study, MORPHEUS offers patients randomized to the active comparator arm a chance to receive experimental therapy in the next line of treatment. With so many arms, there is also a lower potential risk of being randomized to the active comparator. Second, multiple experimental immunotherapy arms can be evaluated side-by-side, avoiding the issue of cross-trial comparisons to decide which regimens are performing best. However, replicating this design would be very difficult with unique agents from small companies with a more limited drug development pipeline.

The feasibility of utilizing polypharmacy to simultaneously manipulate diverse TME pathways remains undetermined. While co-administration of multiple cytotoxic agents targeting activated cancer-cell-intrinsic oncogenic pathways has most frequently generated intolerable toxicity, the side effect profiles of many TME-directed agents appear more benign at first glance. The failure of PEGPH20 + FOLFIRINOX due to excessive toxicity serves as a cautionary tale to investigators that TME-directed agents may have underappreciated side effects subject to amplification by therapeutics with a non-overlapping toxicity profile. Whether sequential (rather than simultaneous) administration of complimentary therapeutics can be successfully utilized to correct multiple immunosuppressive features of the PDAC TME without triggering excessive toxicity may be an important area of future exploration.

It is a legitimate question to ask whether tumors that have simultaneously co-opted host immune tolerance and desmoplasia-producing wound healing programs will run out of compensatory mechanisms even when faced with a bevy of pharmaceutical products designed to manipulate those programs. PDAC is a cancer driven by KRAS activation, and new drug design developments have made blocking KRAS itself possible for the first time. Sotorasib, the first KRAS-directed drug approved, has shown remarkable activity in PDAC patients with KRAS G12C-mutated tumors: a 21% overall response rate with a median treatment duration of 4.1 months in participants who have received a median of two prior therapies [13]. This development marks an exciting landmark in the field and offers a new type of therapy to PDAC patients. However, sotoarasib alone is not providing the kind of sustained clinical benefit that patients are hoping to grasp. In PDAC, perhaps even drugs targeted against its defining oncogenic driver mutation require combination with TME-modulating agents to maximize clinical benefit. Finding the right cocktail of drugs to reprogram PDAC TME will not be easy, but as we fight for each step forward, perhaps the finish line is finally getting closer.

## Figures and Tables

**Figure 1 cancers-14-04209-f001:**
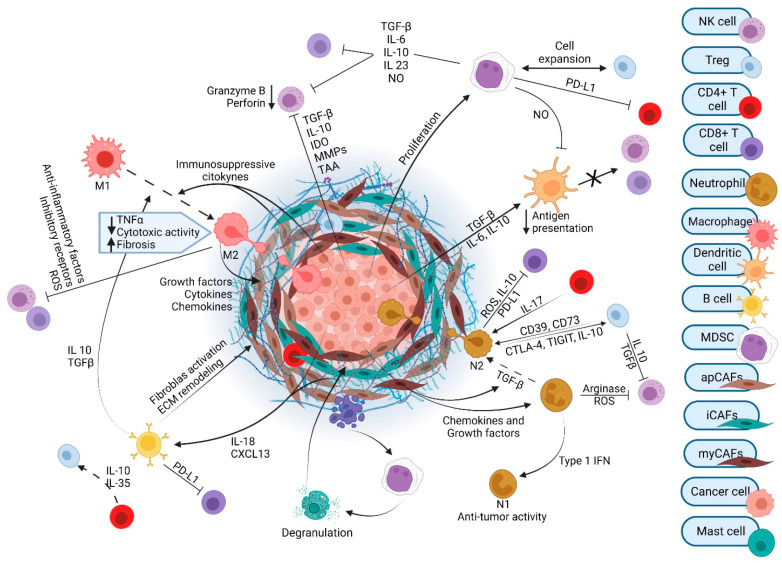
The tumor microenvironment of pancreatic ductal adenocarcinoma.

**Table 1 cancers-14-04209-t001:** List of active clinical trials in PDAC targeting immune cell crosstalk.

Mechanism of Action	NCT	Status	Agent	Combination	Phase	PDAC Patient Population	Results Reported?
**Targeting immune cells**							
**CD40 agonist**	NCT00711191	Comp	Selicrelumab (CP-870,893; RO7009789)	gemcitabine	1	advanced	x
	NCT01456585	Comp	Selicrelumab	Perioperative chemoradiation (gemcitabine)	1	resectable	
	NCT02588443	Comp	Selicrelumab	±GN	1	resectable	x
	NCT03193190	Recr	Selicrelumab	GN + atezolizumab	1/2	advanced	
	NCT03214250	A-NR	Sotigalimab (APX005M)	GN ± nivolumab	1b/2	metastatic	x
	NCT04536077	Recr	CDX-1140	±CDX-301 (FLT3L)	1	resectable	
	NCT02376699	A-NR	SEA-CD40	pembrolizumab ± GN	1	advanced	x
	NCT04888312	Recr	Mitazalimab	mFFX	1/2	metastatic	x
**Oncovirus: trimerized CD40L and 4-1BBL**	NCT02705196	Recr	LOAd703 (delolimogene mupadenorepvec)	GN, atezolizumab	1/2	advanced	x
	NCT03225989	Recr	LOAd703	chemo	1/2	advanced	x
**CXCR4 antagonist**	NCT02179970	Comp	Plerixafor (AMD3100)		1	advanced	
	NCT04177810	Recr	Plerixafor	cemiplimab (anti-PD-1)	2	metastatic	
	NCT02907099	A-NR	Motixafortide (BL-8040)	pembrolizumab	2	metastatic	x
	NCT02826486	A-NR	Motixafortide	pembrolizumab ± Nal-iri/5-FU	2	metastatic	x
	NCT4543071	Recr	Motixafortide	cemiplimab + GN	2		
**CXCL12 antagonist**	NCT03168139	Comp	Olaptesed Pegol (NOX-A12)	pembrolizumab	1/2	metastatic	x
	NCT04901741	NYR	Olaptesed Pegol (NOX-A12)	pembrolizumab, Nal-iri/5-FU or GN	2	MSS metastatic	
**CSF1R inhibitor**	NCT03153410	A-NR	IMC-CS4 (LY3022855)	pembrolizumab, GVAX, cyclophosphamide	1	BR	
	NCT02777710	Comp	Pexidartinib	durvalumab	1	advanced	x
	NCT02713529	Comp	AMG 820	pembrolizumab	1/2	advanced	x
**CD11b agonist**	NCT04060342	A-NR	ADH-503 (GB1275)	pembrolizumab or GN	1/2	advanced	
**STING agonist**	NCT05070247	Recr	TAK-500	±pembrolizumab	1	advanced	
	NCT03010176	Comp	ulevostinag (MK-1454)	±pembrolizumab	1	advanced	x
**CCR2 antagonist**	NCT01413022	Comp	PF-04136309	mFFX	1	BR	x
	NCT02732938	Term	PF-04136309	GN	1/2	metastatic	x
	NCT02345408	Comp	CCX872-B	FFX	1	advanced	x
**CCR2-CCR5 dual antagonist**	NCT03184870	A-NR	BMS-813160	±nivolumab or chemo	1/2	advanced	
**CD73/Adenosine Receptor inhibition**	NCT02503774	A-NR	Oleclumab (MEDI9447)	±durvalumab	1	advanced	x
	NCT03611556	A-NR	Oleclumab	durvalumab and/or chemo	1/2	metastatic	
	NCT03207867	A-NR	Taminadenant (NIR178)	±spartalizumab (anti-PD-1)	2	advanced	
	NCT03549000	A-NR	NZV930 ± taminadenant	±spartalizumab	1	advanced	
	NCT04104672	Recr	Quemliclustat (AB680)	GN ± zimberelimab (anti-PD-1)	1	metastatic	x

Trial status—Comp: completed; Recr: recruiting; A-NR: active, not recruiting; NYR: not yet recruiting; Term: terminated. PDAC Patient—BR: borderline resectable; MSS: microsatellite stable. Treatment—mFFX: modified FOLFIRINOX; GN: gemcitabine + nab-paclitaxel; chemo: standard-of-care chemotherapy; Nal-iri: nanoliposomal irinotecan; 5-FU: 5-fluorouracil.

**Table 2 cancers-14-04209-t002:** Selected list of clinical trials in PDAC targeting stromal components.

Mechanism of Action	NCT	Status	Agent	Combination	Phase	PDAC Patient Population	Results Reported?
**Direct stroma targeting**							
**FAK inhibitor**	NCT03727880	Recr	±defactinib	pembrolizumab	2	resectable	
	NCT04331041	Recr	±defactinib	SBRT	2	locally advanced	
	NCT02546531	Comp	defactinib	pembrolizumab + gemcitabine	1	advanced	x
	NCT02428270	A-NR	GSK2256098	trametinib	2	advanced	
**IL-6 antagonist**	NCT04191421	Recr	Siltuximab	spartalizumab (PD-1)	1/2	metastatic	
	NCT02767557	A-NR	±tocilizumab	GN	2	advanced	
	NCT04258150	Term	tocilizumab	Nivolumab + ipilimumab + XRT	2	advanced	
	NCT03193190	Recr	tocilizumab	GN + atezolizumab	1/2	metastatic	
**VitD receptor agonist**	NCT02030860	Comp	±paricalcitol	GN	1	resectable	
	NCT02930902	A-NR	paricalcitol	pembrolizumab, ±GN	1	resectable	
	NCT02754726	A-NR	paricalcitol	Pembrolizumab + GN + cisplatin		metastatic	x
	NCT03520790	Recr	paricalcitol	GN	2	metastatic	
	NCT03331562	Comp	±paricalcitol	pembrolizumab	2	Metastatic, maint	x
**ATRA**	NCT03307148	Comp	ATRA	GN	1	advanced	
	NCT04241276	A-NR	ATRA	GN	2	locally advanced	
**VEGF inhibition**	NCT03797326	A-NR	lenvatinib	pembrolizumab	2	advanced	
	NCT04887805	Recr	lenvatinib	pembrolizumab	2	advanced, maint	
	NCT05327582	Recr	lenvatinib	durvalumab, nab-paclitaxel	1/2	advanced	
	NCT05303090	Recr	lenvatinib	H-101, tislelizumab	1b	advanced	
	NCT03193190	Recr	bevacizumab	GN, atezolizumab	1/2	metastatic	
**Integrin inhibitor**	NCT00401570	Comp	volociximab	gemcitabine	2	metastatic	x
**Integrin cytotoxin**	NCT05085548	Recr	ProAgio		1	advanced	
**Hyaluronan dissolution**	NCT01453153	Comp	±PEGPH20	gemcitabine	1/2	metastatic	x
	NCT01839487	Comp	±PEGPH20	GN	2	metastatic	x
	NCT02715804	Comp	±PEGPH20	GN	3	metastatic	x
	NCT01959139	A-NR	±PEGPH20	FFX	1/2	metastatic	x
	NCT02910882	Term	PEGPH20	XRT + gemcitabine	2	Locally advanced	
	NCT02241187	Comp	PEGPH20	cetuximab	-	resectable	
	NCT03193190	Recr	PEGPH20	atezolizumab	1/2	metastatic	x
**Angiotensin II receptor blockade**	NCT01821729	A-NR	losartan	FFX + XRT	2	Locally advanced	x
	NCT03563248	Recr	±losartan	FFX + SBRT + surgery, ± nivolumab	2	Resectable, BR or locally advanced	
	NCT04106856	Recr	losartan	Hypofractionated radiation	1	BR or locally advanced	
	NCT05077800	Recr	±losartan	FFX ± elraglusib (9-ING-41; GSK-3β inhibitor)	2	metastatic	
	NCT05365893	Recr	losartan	Paricalcitol + hydroxychloroquine	1	resectable	
	NCT04539808	Recr	losartan	mFFX ± switch to GN followed by capecitabine/XRT	2	Resectable, BR or locally advanced	

Trial status—Comp: completed; Recr: recruiting; A-NR: active, not recruiting; NYR: not yet recruiting; Term: terminated. PDAC Patients–BR: borderline resectable; maint: maintenance. Treatment—mFFX: modified FOLFIRINOX; GN: gemcitabine + nab-paclitaxel; XRT: radiation therapy; SBRT: stereotactic body radiation therapy.

## Data Availability

The data can be shared up on request.

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
