# Peer review of "Clinical Strategies Targeting the Tumor Microenvironment of Pancreatic Ductal Adenocarcinoma"

_cancers, 2022, doi:10.3390/cancers14174209_

Round 1

Reviewer 1 Report (Previous Reviewer 1)

There is still a problem that most clinical trials are ongoing and lack final results. Only basic information is not enough in a review, the author should propose their own suggestions according to the data from clinical trials.

Author Response

We agree that many of the clinical trials cited lack final results. In the newest revision, we have extensively expanded our discussion of those that do have results so that more in depth information is available for readers. In addition, we have added new sections on VEGF inhibition and Angiotensin Receptor blockade. All sections have been enhanced to provide more analysis and the Introduction and Summary now incorporate novel suggestions based upon existing clinical trial data.

Reviewer 2 Report (Previous Reviewer 2)

The authors revised the available strategies targeting the tumor microenvironment of pancreatic ductal adenocarcinoma. 

Point to be considered:

1) The rationale of why the authors came up with this review.

2) The information that motivated the authors to come up with is not precisely available. What are the current caveats, and how do the authors highlight the current research in answering them? If not, they need to address it in future directions.

3) The underlying message here is that more precision and individualized approaches need to be tested in well-designed clinical trials – a challenge, but I would be interested in their perspective of how this might be done.

4) As is now well known, tumors grow and evolve through a constant crosstalk with the surrounding microenvironment, and emerging evidence indicates that angiogenesis and immunosuppression frequently co-occur in response to this crosstalk. Accordingly, strategies combining anti-angiogenic therapy and immunotherapy seem to have the potential to tip the balance of the tumor microenvironment and improve treatment response (please expand and comment)

5) In the frame of point 4) thinking, genetic alterations, especially the K-Ras mutation, carry the heaviest burden in the progression of pancreatic precursor lesions into pancreatic ductal adenocarcinoma (PDAC). The tumor microenvironment is one of the challenges that hinder the therapeutic approaches from functioning sufficiently and leads to the immune evasion of pancreatic malignant cells. Mastering the mechanisms of these two hallmarks of PDAC can help us in dealing with the obstacles in the way of treatment: the signaling pathways involved in PDAC development and the immune system's role in pancreatic cancer and immune checkpoint inhibition as next-generation therapeutic strategy are crucial.  This reviewer personally misses some insights regarding the direct targeting of the involved signaling molecules and the immune checkpoint molecules, along with a combination with conventional therapies, have reached the most promising results in pancreatic cancer treatment (please refer to PMID: 33918146 and expand the introduction/discussion sections).

6) The authors need to highlight what new information the review is providing to enhance the research in progress

Author Response

Point to be considered:

1) The rationale of why the authors came up with this review.

Additional text has been added to the last paragraph of the Introduction and Summary & Perscpective sections to better enunciate to the reader the rationale and aims of this article.

2) The information that motivated the authors to come up with is not precisely available. What are the current caveats, and how do the authors highlight the current research in answering them? If not, they need to address it in future directions.

Additional text has been added to the last paragraph of the Introduction and Summary & Perspective sections to better enunciate the current caveats in the field.

3) The underlying message here is that more precision and individualized approaches need to be tested in well-designed clinical trials – a challenge, but I would be interested in their perspective of how this might be done.

This perspective has been added to the Summary and Perspectives section.

4) As is now well known, tumors grow and evolve through a constant crosstalk with the surrounding microenvironment, and emerging evidence indicates that angiogenesis and immunosuppression frequently co-occur in response to this crosstalk. Accordingly, strategies combining anti-angiogenic therapy and immunotherapy seem to have the potential to tip the balance of the tumor microenvironment and improve treatment response (please expand and comment)

Thank you for this excellent suggestion. A new sub-section reporting on use of anti-VEGF drugs has been added to Section 5. This new text details past failures of these drugs in combination with chemotherapy and erlotinib and outlines the field’s resurgent interest in these therapies given exciting results for combination with anti-PD(L)1 therapies that have been seen in other tumor types.  In addition, we have included a section on losartan treatment for improving vascular dynamics.

5) In the frame of point 4) thinking, genetic alterations, especially the K-Ras mutation, carry the heaviest burden in the progression of pancreatic precursor lesions into pancreatic ductal adenocarcinoma (PDAC). The tumor microenvironment is one of the challenges that hinder the therapeutic approaches from functioning sufficiently and leads to the immune evasion of pancreatic malignant cells. Mastering the mechanisms of these two hallmarks of PDAC can help us in dealing with the obstacles in the way of treatment: the signaling pathways involved in PDAC development and the immune system's role in pancreatic cancer and immune checkpoint inhibition as next-generation therapeutic strategy are crucial.  This reviewer personally misses some insights regarding the direct targeting of the involved signaling molecules and the immune checkpoint molecules, along with a combination with conventional therapies, have reached the most promising results in pancreatic cancer treatment (please refer to PMID: 33918146 and expand the introduction/discussion sections).

We have included an additional paragraph in the Introduction which discusses the molecular landscape of PDAC and how this relates to precision targeting. Besides this, we have already addressed use of therapies targeted against oncogenic drivers with our previous revision (please see the first paragraph of Section 3). These types of therapies are not a main focus of our review as this has been extensively detailed elsewhere as pointed out by the reviewer.

6) The authors need to highlight what new information the review is providing to enhance the research in progress

We hope that the reviewer agrees that the extensive revisions that have been made successfully serve this purpose.

Round 2

Reviewer 2 Report (Previous Reviewer 2)

This is a revised version. The authors have clarified several of the questions I raised in my previous review. Most of the major problems have been addressed by this revision.

Author Response

We are delighted that our newest revision has addressed your concerns.

This manuscript is a resubmission of an earlier submission. The following is a list of the peer review reports and author responses from that submission.

Round 1

Reviewer 1 Report

In this study, the author summarized clinical strategies targeting the tumor microenvironment of pancreatic ductal adenocarcinoma. However,most clinical trials are ongoing and without final results, thus the author only listed basic information and cannot provide digital support and guidance for clinical applications. Considering the important role of immune cells, more details of immunotherapy should be discussed in this study, including immune checkpoint inhibitors, tumor vaccines, adoptive T cell therapy, oncolytic viruses, etc. As such, the significance of the work is limited.

Author Response

As per the recommendations of Reviewer 2, we have clarified the scope of our manuscript in the Introduction. Specifically, we are interested in therapeutics designed to directly target TME components. We believe that discussions of adoptive T cell therapies, cancer vaccines or oncolytic viruses are outside the scope of our manuscript, which was not intended as a general review of PDAC-directed immunotherapies. We have not included extensive information about these topics, however, we have revised the first paragraph of section 4 to mention that these strategies are actively being tested in PDAC, provided relevant references of key studies, and referred the reader to other articles which could provide more extensive information.

Studies of immune checkpoint inhibitors in PDAC are described in the first paragraph of Section 4. Additional information about the mechanism of action for CTLA-4 and PD-(L)1 therapeutics has been added to this section in response to the reviewer’s concern.

We agree that many of the clinical trials presented are ongoing and lack final results. Nevertheless, we believe that this information can be helpful to basic and translational researchers interested in studying combination treatments who may be unfamiliar with which drugs have advanced to clinical trials and what preliminary results are available from those trials (ie. which strategies might still be feasibly translated to clinic if successful in the pre-clinical setting). We also think it is important for helping clinicians designing new clinical trials to appreciate the current breadth of this field.

Reviewer 2 Report

Nebojsa Skorupan et al. uncovered some interesting aspects of PDAC TME targeting.

Points to be addressed:

1) The rationale of why the authors came up with this review.

2) What is the information that is not exactly available that motivated the authors to come up with this information. What are the current caveats and how do the authors highlight the current research in answering them? If not they need to address in future directions.

3)This review personally misses some aspects regarding genetic alterations, especially the K-Ras mutation, carrying the heaviest burden in the progression of pancreatic precursor lesions into PDAC. The tumor microenvironment is one of the challenges that hinder the therapeutic approaches from functioning sufficiently and leads to the immune evasion of pancreatic malignant cells. Mastering the mechanisms of these two hallmarks of PDAC can help us in dealing with the obstacles in the way of treatment. In this review, we have analyzed the signaling pathways involved in PDAC development and the immune system's role in pancreatic cancer and immune checkpoint inhibition as next-generation therapeutic strategy. The direct targeting of the involved signaling molecules and the immune checkpoint molecules, along with a combination with conventional therapies, have reached the most promising results in pancreatic cancer treatment (please expand referring to PMID: 33918146);

4) The authors need to highlight what new information the review is providing to enhance the research in progress.

5) The underlying message here is that more precision and individualized approaches need to be tested in well designed clinical trials – a challenge, but I would be interested in their perspective of how this might be done.

Author Response

1, 2 & 4) We have augmented the Introduction and Summary & Conclusions sections to address our rationale for the review, to better define the scope of our work, to examine caveats in the field of TME-directed therapeutics, and to speculate on the future of the field.

3) We agree that we neglected to discuss the role of mutated KRAS in driving development of the TME during the tumorigenesis process and have added a new first paragraph to Section 3 discussing this. In this review, we are examining therapeutics which directly target TME components rather than targets within cancer cells (and have clarified this scope in the Introduction). Therefore, we have not included a discussion of anti-KRAS or MEK/ERK pathway drugs in combination with immunotherapy in the main body of our manuscript.

5) We agree that the design of clinical trials in this field is tricky and have expounded on the importance of combination studies and the difficulties of rationally designing them.

Thank you for your excellent suggestions. 

Reviewer 3 Report

This is a well-summarized and comprehensive review about PDAC microenvironment and current strategies to target microenvironment factors for better cure of PDAC. The manuscript are generally well-prepared. I would only recommend some minor revision before publication:

1) page 2 line 53: author claimed desmoplasia contains aSMA as one of the ECM factors. I think it is a typo here. aSMA is a intracellular protein.

2) Figure 1 is a gigantic and complex drawing. I understand the authors have put great effort in making this figure. I would still recommend some adjustments: a) uniform the text format. 'Degranulation' is bolded/bigger but others are not. b) Are there evidences showing CAF subtypes surrounding the tumor differs based on distance? I would imagine a more heterogenous distribution of CAF subtypes instead of 'layer-by-layer' separation.

Author Response

  1. This error including a-SMA has been corrected. Thank you for helping us to improve this manuscript.
  2. Figure 1 means to show the complex interplay between cancer cells and TME components. We have edited the figure text and localization of CAFs as suggested. Only one original publication was found describing the localization of CAFs in PDAC. Ohlund et al (new reference 16) identified a CAF subpopulation with elevated expression of α-smooth muscle actin (αSMA) located immediately adjacent to neoplastic cells in mouse and human PDA tissue. In co-cultures authors revealed another distinct subpopulation of CAFs, located more distantly from neoplastic cells, which lacked elevated αSMA expression and secreted inflammatory mediators (among them IL-6). These cells were later named iCAFs.